# BCD Nets: Scalable Variational Approaches for Bayesian Causal Discovery

**Chris Cundy**[1]   **Aditya Grover**[2,3]   **Stefano Ermon**[1]

[1]Department of Computer Science, Stanford University
[2]Facebook AI Research
[3]University of California, Los Angeles

{cundy, ermon}@cs.stanford.edu
adityag@cs.ucla.edu

## Abstract

A structural equation model (SEM) is an effective framework to reason over causal relationships represented via a directed acyclic graph (DAG). Recent advances have enabled effective maximum-likelihood point estimation of DAGs from observational data. However, a point estimate may not accurately capture the uncertainty in inferring the underlying graph in practical scenarios, wherein the true DAG is non-identifiable and/or the observed dataset is limited. We propose Bayesian Causal Discovery Nets (BCD Nets), a variational inference framework for estimating a *distribution* over DAGs characterizing a linear-Gaussian SEM. Developing a full Bayesian posterior over DAGs is challenging due to the the discrete and combinatorial nature of graphs. We analyse key design choices for scalable VI over DAGs, such as 1) the parametrization of DAGs via an expressive variational family, 2) a continuous relaxation that enables low-variance stochastic optimization, and 3) suitable priors over the latent variables. We provide a series of experiments on real and synthetic data showing that BCD Nets outperform maximum-likelihood methods on standard causal discovery metrics such as structural Hamming distance in low data regimes.

## 1 Introduction

One of the key uses of statistical methods is learning causal relationships from observed data a.k.a. *causal discovery* [39]. Causal models allow us to forecast the effects of interventions and counterfactuals in several real-world domains, such as economic policy [57] and medicine [47]. Although early approaches to statistical inference emphasised that 'correlation is not causation' [15], it has since been shown that for certain families of data-generating processes, it is indeed possible to infer causal relationships from purely observational data [37, 29].

One such widely studied data-generating process is the linear-Gaussian structural equation model (SEM) [37], where the causal relationships between the random variables in the model can be represented via a weighted directed acyclic graph (DAG). The value of any variable in the DAG of a linear-Gaussian SEM is given by a linear combination of the values of its parent nodes and additive noise. For causal discovery, naive Monte-Carlo sampling or enumeration of the possible DAGs quickly becomes intractable, since the number of possible DAGs over a model grows superexponentially with the number of variables [18]. A variety of methods have been developed over the years to efficiently sample or optimize over DAGs [11, 53, 51, 45]. For example, a recent line of work scales to high dimensions by maximizing the likelihood of the model (MLE) over a set of continuous relaxations of adjacency matrices using gradient-based methods and specialized DAG regularization terms. [62, 61, 35, 59].

35th Conference on Neural Information Processing Systems (NeurIPS 2021).

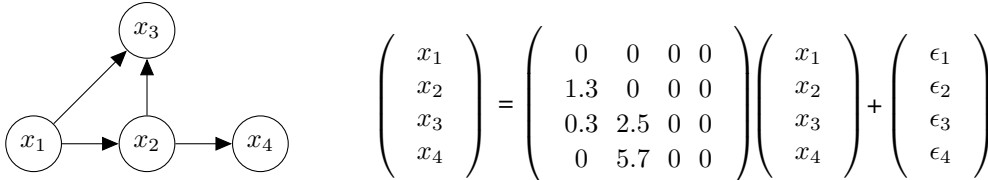

Figure 1: A Structural Equation Model (SEM). **Left:** DAG with 4 nodes. **Right:** Linear-Gaussian SEM (with $\epsilon \sim \mathcal{N}(0, \Sigma)$)

However, the majority of the aforementioned works for scalable causal discovery in linear-Gaussian SEMs focus on recovering *point estimates* for the underlying DAG via MLE. In many practical scenarios, however, a point estimate fails to reflect the uncertainty in inferring the underlying DAG. This includes scenarios where the true DAG is non-identifiable given (infinite) observational data as well as practical limitations due to an imperfect optimization algorithm, model mismatch, or simply a finite dataset. In any of the above scenarios, it is desirable to obtain an explicit *posterior distribution* over the unobserved DAG instead of a single point estimate [19]. Causal inference is increasingly being applied in situations with important real-world consequences, where such a Bayesian estimation procedure could be useful to sample and reason over alternative generative mechanisms for observed data.

To achieve this goal, we propose Bayesian Causal Discovery Nets (BCD Nets), an algorithmic framework for Bayesian causal discovery in linear-Gaussian SEMs based on modern variational inference [6]. Classical approaches to Bayesian causal discovery struggle in higher dimensions [23, 16], with limited improvements via Monte Carlo approximations [22, 58]. In order to scale our variational method to high dimensions, we address several design challenges. First, we describe an expressive variational family of factorized posterior distributions over the SEM parameters (edge weights and noise variance) using deep neural networks. The factorization exploits the decomposition of DAGs into triangular matrices and permutations for specifying the distribution over edge weights and node orderings respectively. Further, for low-variance stochastic optimization of the variational objective, we exploit recent advances in modeling and reparametrizing distributions over permutations via continuous relaxations [31, 28]. Finally, we employ a horseshoe prior [7] on the edge weights, which promotes sparsity.[1]

We evaluate BCD Nets for causal discovery on a range of synthetic and real-world data benchmarks. Our experiments demonstrate that with finite datasets, there is considerable uncertainty in the inferred posterior over DAGs. Using BCD Nets, we are able to effectively quantify the uncertainty and significantly outperform competing estimators [35, 58] on the standard Structured Hamming Distance metric, especially in low data regimes.

## 2 Preliminaries

### 2.1 Linear-Gaussian Structural Equation Models

A structural equation model (SEM) is a collection of random variables $x_1, \ldots, x_d$ associated with a directed acyclic graph (DAG) $G$ with $d$ nodes [37]. The SEM consists of a series of equations $x_j = f_j(\text{Pa}(x_j), \epsilon_j)$, where $\text{Pa}(x_j)$ gives the values of the parents of the $j$th node in $G$ and $\epsilon_j$ is a noise variable. For a linear-Gaussian SEM, the equations $f_j$ are linear and the noise $\epsilon_j$ is additive Gaussian. Considering $X = [x_1, \ldots, x_d]$ as a vector in $\mathbb{R}^d$ and $W$ as the weighted adjacency matrix of $G$, we can see that $X$ must satisfy $X = W^\top X + \epsilon$, where $\epsilon \sim \mathcal{N}(0, \Sigma)$ is the additive noise vector and $\Sigma = \text{diag}\left\{\sigma_1^2, \ldots, \sigma_d^2\right\}$ is a diagonal noise covariance matrix. We will denote the setting when all noise variances are equal i.e., $\sigma_1 = \ldots = \sigma_d = \sigma$ as an "equal variance" setting and "non-equal variance" otherwise. Figure 1 shows an illustration. Linear SEMs have been used to model microarray data [38] and protein pathways [13], among many other systems.

---

[1]Code is available at github.com/ermongroup/BCD-Nets

## 2.2 Causal Discovery

Given some dataset $X_1^n = \{X_1, X_2, \ldots, X_n\}$ drawn i.i.d. from a linear-Gaussian SEM, we are interested in inferring the adjacency matrix $W$ and the diagonal entries of the covariance matrix $\Sigma$.

**Maximum Likelihood Estimation (MLE)**  This approach obtains a point estimate for $W$ and $\Sigma$ by maximizing the likelihood of the dataset $X_1^n$. Using the fact that $W$ is constrained to be a DAG, it can be shown that $I - W^\top$ is always invertible.[2] Hence, we can rearrange to obtain $X = (I - W^\top)^{-1}\epsilon$, obtaining $X$ as the product of a matrix and a Gaussian vector. It follows that $X$ itself has a Gaussian distribution, $X \sim \mathcal{N}(0, \Theta^{-1})$, with precision matrix $\Theta = (I - W)\Sigma^{-1}(I - W)^\top$. This gives a joint log-likelihood over the dataset $X_1^n$ as

$$\log p(X_1^n; \Sigma, W) = \frac{n}{2}\left(\log \det \Theta - d\log(2\pi)\right) - \frac{1}{2}\sum_{i=1}^{n} X_i^\top \Theta X_i, \tag{1}$$

where $X_i$ runs over the $n$ points in the dataset and $\Theta$ is given above. The space of DAG adjacency matrices $W$ is characterised by the acyclicity constraint. When viewed as an subset of the space of all adjacency matrices ($\mathbb{R}^{d \times d}$), this is a piecewise linear manifold with a number of facets that grows super-exponentially in the dimension (roughly as $d!2^{d^2/2}$), rendering any approach based on enumeration over all DAGs intractable. Many approaches have been proposed to scale causal discovery via MLE to high-dimensional data. This includes recent approaches which relax the DAG constraint to a larger, continuous set of adjacency matrices, such as $\mathbb{R}^{d \times d}$ in [62] or the Birkhoff polytope of doubly-stochastic matrices in [5], which enables the use of gradient-based optimization.

However, there are a number of challenges with using point estimators for causal discovery. Fundamentally, the true SEM parameters are identifiable from observational data only under specific conditions. In fact, the map between the the data distribution parameters $\Theta$ and the SEM parameters $\{W, \Sigma\}$ may not be bijective even in the limit of infinite data and an oracle optimizer. Theoretical results on identifiability in linear-Gaussian SEMs currently hold only under restricted settings; notably, this only includes the equal variance setting [38]. Further, even if the noise variances are equal in the ground-truth SEM, we are limited in practice by the finite size of the dataset and MLE could still converge to an incorrect solution. Finally, such point estimators are typically not robust to potential misspecifications in the likelihood model, e.g., a non-Gaussian noise model, non-linear cause-effect relationships, etc. For the above scenarios, it is useful to characterize the uncertainty in estimating the SEM parameters to guide downstream analysis. For example, in medical applications giving a distribution of possible causal pathways, with quantified uncertainty, could be much more useful than a point estimate of the most likely pathway.

**Bayesian Estimation.**  In contrast to point estimators, Bayesian methods explicitly characterize the uncertainty in the estimated parameters [19]. That is, we treat the unknown SEM parameters $\{W, \Sigma\}$ as random variables associated with a prior distribution $p(W, \Sigma)$. The likelihood model $p(X_1^n|\Sigma, W)$ follows the same expression as the RHS in equation (1). Given the prior and the likelihood, we obtain a posterior distribution $p(W, \Sigma|X_1^n)$ over $\{W, \Sigma\}$ via Bayes rule, which quantifies the uncertainty in estimating the SEM parameters. As long as the likelihood model is well-specified and the prior includes the ground-truth SEM parameters in its support, as the dataset size increases the posterior will concentrate around a set of SEM parameters. In the equal variance case these will be the true SEM parameters. In the non-equal variance case the posterior will concentrate on the set of DAG parameters quasi-equivalent to the ground truth as defined in [35]. These are the set of DAG parameters which generate data with the same covariance as the ground truth, and are hence indistinguishable based on data alone. We discuss this further in the appendix, Section F.

The key challenge in Bayesian estimation is tractable computation of the posterior distribution in high-dimensional spaces. With the exception of specific prior and likelihood families e.g., conjugate distributions, computing the posterior is typically intractable. In the next section, we present a variational framework for scalable approximation of the posterior for Bayesian causal discovery.

---

[2]Writing $W = PLP^\top$ from Section 3, $I - W^\top = (I - PL^\top P^\top) = P(I - L^\top)P^\top$. Now $(I - L^\top)$ is upper-triangular with unit diagonal, so $\det(I - L^\top) = 1$. Thus $(I - W^\top)$ has unit determinant, so is invertible.

# 3 Causal Discovery via Bayesian Causal Discovery Nets

As discussed previously, we are interested in learning the posterior distribution $p(W, \Sigma \mid X_1^n)$ over the unknown SEM parameters $\{W, \Sigma\}$ given an observed dataset $X_1^n$. Unlike point estimators, such a posterior distribution will allow us to quantify the uncertainty in estimation. Our framework, Bayesian Causal Discovery Nets (BCD Nets) allows us to tractably estimate this posterior.

As a first step, our approach involves parametrizing the adjacency matrix $W$ as the product of a permutation matrix $P$ and a strictly lower-triangular matrix $L$, so that $W = PLP^\top$. In graphical terms, $L$ is a weight matrix for a canonical DAG with a fixed ordering, while pre- and post-multiplication by $P$ and $P^\top$ modifies the ordering of nodes. $L$ is parameterised by a vector of weights $l \in \mathbb{R}^{d(d-1)/2}$, and the constraint that $P$ is a permutation ensures that $W$ is the adjacency matrix of a DAG.

Our goal is to obtain the posterior distribution $p(P, L, \Sigma | X_1^n)$. Due to the intractable partition function, we cannot directly compute the posterior. We turn to variational inference to deliver a tractable approximation to the posterior [24]. The key idea here is to cast inference as an optimization problem, wherein we approximate the true posterior with a tractable family of distributions $q_\phi(P, L, \Sigma)$ parameterized by $\phi$ and optimize these parameters $\phi$ to minimize the KL divergence between the approximate and true posterior distributions:

$$
\begin{aligned}
& D_{\text{KL}}\left(q_\phi(P, L, \Sigma) \| p(P, L, \Sigma \mid X_1^n)\right) \\
&= \underbrace{-\mathbb{E}_{(P,L,\Sigma)\sim q_\phi}\left[\log p(X_1^n | P, L, \Sigma) - \log \frac{q_\phi(P, L, \Sigma)}{p(P, L, \Sigma)}\right]}_{\text{ELBO}(\phi)} + \log p(X_1^n).
\end{aligned}
\tag{2}
$$

Hence, minimizing the KL divergence above corresponds to maximizing the evidence lower bound (ELBO) w.r.t. variational parameters $\phi$. With a sufficiently expressive variational family from which to choose $q$, maximizing the ELBO recovers the true posterior as $q_\phi(P, L, \Sigma) = p(P, L, \Sigma | X_1^n)$.

In practice, we face important modeling choices which have a substantial impact on the quality of the posterior obtained, as well as the difficulty of optimizing the ELBO. These include the variable ordering to use when factorizing $q_\phi(P, L, \Sigma)$ using the chain rule, choice of variational family for the individual (conditional) factors, as well as the prior distribution $p(P, L, \Sigma)$. We discuss these algorithmic design choices next.

## 3.1 Factorization of Approximate Posterior

Approaches to variational inference with multiple sets of latent variables often use a mean-field factorization [24], in our case corresponding to $q_\phi(P, L, \Sigma) = q_\phi(P)q_\phi(L)q_\phi(\Sigma)$. This mean-field approach can often simplify the optimization of the ELBO, but severely limits the expressiveness of the approximate posterior. For example, consider a two-dimensional linear-Gaussian SEM with non-equal variances (and therefore has non-uniquely identifiable parameters). Under infinite data, the posterior density concentrates on the two (observationally undistinguishable) maximum-likelihood solutions: an edge $x_1 \rightarrow x_2$ with some weight $l_1$ and an edge $x_1 \leftarrow x_2$ with another weight $l_2$. The posterior concentrates to a bimodal distribution, with density around the region $(l_1, P_1)$ and around $(l_2, P_2)$. A mean-field factored posterior $q_\phi(L)q_\phi(P)$ cannot represent this correlated density. Empirically we observe that such a factored posterior leads to a worse ELBO, illustrated in ablation experiments in Section 5.6.

For BCD Nets, we use a factorization $q_\phi(P, L, \Sigma) = q_\phi(P|L, \Sigma)q_\phi(L, \Sigma)$, sampling $L$ and $\Sigma$ jointly first, then conditionally sampling $P$ based on these values, using a neural network to learn the parameters of the conditional distribution $q_\phi(P|L, \Sigma)$. This leads to an ELBO

$$
\mathbb{E}_{(L,\Sigma)\sim q_\phi}\left[\mathbb{E}_{P\sim q_\phi(\cdot|L,\Sigma)}\left[\log p(X_1^n | P, L, \Sigma) - \log \frac{q_\phi(P|L,\Sigma)}{p(P|L,\Sigma)}\right] - \log \frac{q_\phi(L,\Sigma)}{p(L,\Sigma)}\right]
\tag{3}
$$

## 3.2 Variational Families

An important design choice in variational methods is the variational family used. A distribution $q_\phi$ over latents $z$ used in ELBO optimization must support two operations: drawing a sample $z \sim q_\phi$, and computing $\log q_\phi(z) - \log p(z)$. Depending on the prior distribution $p$, it may be additionally

possible to compute the term $\mathbb{E}_{z \sim q_\phi} [\log q_\phi(z) - \log p(z)] = D_{\text{KL}}[q_\phi, p]$ in closed form, possibly reducing variance compared to Monte Carlo estimates [43]. It is also desirable that sampling $z \sim q_\phi$ can be written as $g_\phi(\gamma), \gamma \sim q_0$, i.e. that $z$ is obtained by sampling from a fixed distribution $q_0$ and transformed through a parameterized, differentiable sampling path $g_\phi$. This lets us use pathwise gradient estimators [34], which typically have lower variance than the score-function alternatives [60].

### 3.2.1 Distribution over Weights & Noise Variances

We consider two different variational families for the distribution over weights and noise variances, $q_\phi(L, \Sigma)$, depending on the modeling assumptions over the noise. Under the equal variance modeling assumption, we parameterize our variational family as a (diagonal covariance) normal distribution, with $\phi$ directly encoding the mean and variance of the $d(d-1)/2$ random variables for $L$ and the single random variable for $\sigma$. We use this simple distribution since we expect the posterior over $L$ to be relatively unimodal in the identifiable equal variance case.

In the non-equal variance case, and in the experiments using real-world data, we use a normalizing flow [41] for $q_\phi(L, \Sigma)$. We expect that in this case the distribution over $L$ could be much more complicated, and so a more expressive density model is desirable. We use continuously indexed normalizing flows [10], a recently-developed family of flows offering good performance on multimodal densities. As desired, both the normal distribution and flows have pathwise gradient estimators.

### 3.2.2 Distribution over permutations

Since the set of $d$-dimensional permutation matrices $\mathcal{P}_d$ is discrete and its size scales combinatorially with $d$, it is challenging to specify a variational family of distributions over $P \in \mathcal{P}_d$ that permits both density estimation and sampling for low-variance stochastic optimization of the ELBO objective in equation (3). Since permutations are discrete, pathwise gradient estimators do not exist. Hence, we consider relaxations to distributions over permutations. Our base distribution is the Boltzmann distribution over $\mathcal{P}_d$, parametrised by $T \in \mathbb{R}^{d \times d}$ with probability $P_T(P) \propto \exp\langle T, P \rangle$ for $P \in \mathcal{P}_d$.

**Density estimation.** Computing the partition function for the Boltzmann distribution, $\sum_{P \in \mathcal{P}_d} P_T(P)$ involves an expensive enumeration and is therefore intractable to evaluate in high dimensions. In order to approximate the partition function, we follow [28] in noting that the partition function is equal to the matrix permanent $\text{perm}(\exp T)$. This can in turn be approximated tractably via the Bethe permanent, denoted as $\text{perm}_B(\exp T)$. The Bethe permanent is known to satisfy $\log \text{perm}\, T - \frac{d}{2} \log 2 \leq \log \text{perm}_B T \leq \log \text{perm}\, T$, so that the density will be over-estimated, by no more than a factor of $\frac{d}{2} \log 2$ [3]. We refer the reader to appendix C in [32] for an efficient implementation of the Bethe permanent estimator based on message passing.

**Pathwise Gradient Estimation.** Exact sampling from the Boltzmann distribution is challenging for similar reasons as exact density estimation. Moreover, even tractable low-rank approximations to the Boltzmann distribution based on Gumbel-Matching distributions [55] are not useful as they involve non-differentiable operations and so cannot be used to derive a pathwise gradient estimator. Instead, we use a relaxation to the Gumbel-Matching distribution, the Gumbel-Sinkhorn distribution [31].

To draw a sample from the Gumbel-Sinkhorn distribution with parameters $T$, we calculate $S((T + \gamma)/\tau)$, with $S$ the Sinkhorn operator [50], $\gamma$ a matrix of i.i.d standard Gumbel noise and $\tau$ a temperature hyperparameter. $S(T)$ returns the fixed point obtained from repeated row and column normalization, starting from the elementwise $\exp$ of $T$. In the limit of an infinite number of iterations, this returns a doubly stochastic matrix. As $\tau$ approaches zero, the samples approach samples from $\mathcal{P}_d$, with a distribution given by the Gumbel-Matching distribution. A proof of this fact is given in the appendix of [31]. As the Sinkhorn algorithm is a differentiable function of standard Gumbel noise, we can use a pathwise gradient estimator of gradients involving samples from the Gumbel-Sinkhorn distribution. Additional implementation details for our Sinkhorn approach are in the appendix, Section B.

## 3.3 Prior Distributions

A key aspect of any probabilistic model is the choice of prior distribution for the unknown parameters. The prior incorporates domain knowledge into the problem. Moreover, specific choices of prior can be computationally friendly.

**Gaussian Prior.** We show in the appendix (Section A) that if we choose the prior over edge weights to be an isotropic Gaussian, we can analytically marginalize out the weights, only requiring the distribution over $P$ to characterise the full posterior. Once we have $P$, it is straightforward to obtain $L$, since it is a regression problem which can be solved tractably [54]. Although it is very convenient to avoid modeling a distribution over $L$, in practice we find that the Gaussian generative assumption on the weights is not particularly useful for datasets which we would like to analyse, for which the underlying DAGs are typically sparse. A sample from the distribution over DAGs with Gaussian edge weights will likely have many large- or moderate-weight edges.

**Laplace Prior.** Previous work finding the maximum-likelihood solution of equation (1) has added a term $\lambda\|W\|_1$ penalizing the $L_1$ norm of the adjacency matrix [35, 62], which can be interpreted as imposing an isotropic Laplace prior. The Laplace prior is known to induce sparsity in the posterior [54], but there is generally no way to choose the regularization coefficient $\lambda$ without cross-validation.

**Horseshoe Prior.** Given the limitations of the above choices of priors, we instead propose to use a horseshoe prior on $L$. The horseshoe prior has a sharp peak at zero and relatively flat tails which tend to induce sparsity in the posterior while not significantly penalizing larger coefficients [7]. Mathematically, a variable $\beta_i$ has a horseshoe distribution if it is the result of first drawing a random variable $\lambda_i \sim C^+(0,1)$ from a half-Cauchy distribution, then sampling $\beta_i \sim \mathcal{N}(0, \lambda_i^2\eta^2)$. The parameter $\eta$ can be adjusted to encode the prior belief on the degree of the DAG generating the data. Based roughly on [40], we suggest a rule of thumb of setting $\eta \approx \rho/(d\sqrt{n})$, with $\rho$ the prior belief of the average degree of the DAG. This results in a sparsity prior that penalises more stringently with more data, similarly to the BIC score [46].

### 3.4 Overall Approach

Incorporating all the above design choices, we obtain our overall algorithm for BCD Nets. The pseudocode is shown in Algorithm 1. Here, $q_\phi(L, \Sigma)$ is parameterized as either a normal distribution or a normalizing flow depending on the modeling assumption of equal or non-equal variances. The distribution $q_\phi(P|L, \Sigma)$ is a Gumbel-Sinkhorn relaxation of the Gumbel-Matching distribution over permutations, parameterized by a function $h_\phi$ conditioned on $L, \Sigma$. For the function $h_\phi$, we use a simple two-layer multi-layer perceptron. For stochastic optimization w.r.t. this distribution, we additionally find it useful to use the straight-through gradient estimator [4]. This means that on the forward pass of the backpropagation algorithm, we obtain the $\tau \to 0$ limiting value of the Sinkhorn relaxation using the Hungarian algorithm [27], giving a hard permutation $P$. On the backward pass the gradients are taken with respect to the finite-$\tau$ doubly-stochastic matrix $\tilde{P}$. The prior is given by $p(P, L, \Sigma) = p(P)p(L)p(\Sigma)$ where $p(L)$ is a horseshoe prior, $p(P)$ is a uniform prior over permutations and $p(\Sigma)$ is a relatively uninformative Gaussian prior on $\log \Sigma$.

---

**Algorithm 1:** Bayesian Causal Discovery Nets (BCD Nets)

---

**Input :** data $X_1^n$, Gradient-based optimizer `step`, temperature hyperparameter $\tau$
Initialize parameterized distribution $q_\phi$, neural network $h_\phi(L, \Sigma)$
**while** *not converged* **do**
    Draw $L, \Sigma \sim q_\phi(L, \Sigma)$
    Compute logits $T = h_\phi(L, \Sigma)$
    Draw $\gamma \in \mathbb{R}^{d \times d}$ i.i.d from standard Gumbel
    Compute soft $\tilde{P} = S((T + \gamma)/\tau)$, hard $P = \text{Hungarian}(\tilde{P})$
    Compute $g = \nabla_\phi [\text{ELBO}(\phi)]$ from equation (3) with sampled $P, L, \Sigma$, using $P$ in the
      forward pass and $\tilde{P}$ in the backward pass
    Update $\phi$ via `step` using gradient $g$
**end**

---

## 4 Related Work

**Structure Learning for Bayesian Networks**: The field of Bayesian structure learning investigates how to infer the structure of Bayesian networks from data. Learning the structure of graphical models from data is known to be NP-hard [8]. Nevertheless, two main families of approaches have

been developed to tackle this problem. The first are constraint-based approaches [52], aiming to infer conditional independences from the data and learn a structure consistent with these independences.

The second family of approaches are score-based methods [30, 26], which assign a score to a proposed DAG based on the goodness-of-fit to the data, and search over the space of DAGs to find a structure which maximizes the score. The combinatorial size and discrete nature of the set of possible DAGs in $d$ dimensions present the main difficulties to this approach. Several competing scores have been proposed [46, 1], which generally have an either implicit or explicit term penalizing complexity to avoid overfitting to the data. Recent work has focussed on developing approximate algorithms that can return a high-scoring graph with probabilistic guarantees [36] and on exact methods for graphs under structural assumptions, such as an upper bound on the number of parents of a node [12, 45], similar to the classic Chow-Liu algorithm [9]. Using these techniques it is possible to search over all graphs where nodes have at most two parents for dimension $d > 1000$ [48].

Another set of works improve sampling from distributions over DAGs, e.g. approaches sampling over orderings instead of over DAGs [53]. Given a particular network structure, there are several approaches to learning DAG edge weights, even in the presence of latent variables [33].

**Continuous Relaxations for Structure Learning**: The problem of learning the structure of probabilistic graphical models on *undirected* graphs can be formulated as a convex optimization problem [17], which allows fast inference even for large graphs. However, the acyclicity constraint for learning directed graphs means continuous optimization cannot be applied directly. In recent years, relaxation-based approaches to structure learning have emerged, based on the idea of learning the adjacency matrix corresponding to a DAG via optimization over a matrix $W \in \mathbb{R}^{d \times d}$, instead of in the set of adjacency matrices corresponding to DAGs. This approach was introduced in Zheng et al. [62], with the least-squares loss $\|X - W^\top X\|_F^2$ used to measure the fit of the data to $W$. To encourage $W$ to approximately form an adjacency matrix, a penalty $h(W)$ was added to the loss, where $h(W) = 0$ only if $W$ is a DAG. Originally the penalty $h(W) = \operatorname{Tr} e^{W \odot W}$ was used, and subsequently different penalties have been proposed [61].

Particularly of interest to us is GOLEM [35], which specifically investigates relaxed optimization approaches for the linear-Gaussian SEM that we consider. They point out the similarity of the squared loss to the log-likelihood in equation (1), and show that the additional term $\log \det(I - W)$ also serves as a penalty for non-DAG $W$. This allows optimization over $W$ to maximize the likelihood without the augmented Lagrangian optimization procedure used in Zheng et al. [62].

**Learned Orderings**: There are strong connections between our SEM setting and the problem of learning an ordering for autoregressive models. An SEM is similar to an autoregressive model [2] in the sense that the value at the $i$th index depends on (possibly) all of the values before $i$. It has been observed that certain orderings for autoregressive models can lead to higher likelihood [20], and subsequent work [25, 28] has developed methods to find the orderings which lead to the best performance from the autoregressive model. The closest to our work is Li et al. [28], which uses latent permutations to learn an ordering for an autoregressive model. However, this approach doesn't allow reparameterised gradients due to the hard assignment for the autoregressive order, and the REINFORCE [60] gradient estimator is used instead of lower-variance reparameterised gradients.

## 5   Experiments

In this section we study the empirical performance of our method. On synthetic data, we show that our distributional approach outperforms baselines, including the maximum likelihood approach, in the low-data regime in terms of edges identified correctly. On a real-word protein dataset [44], we also identify more edges correctly compared to the maximum-likelihood method. On a toy causal inference problem, we outperform competing methods. Finally, we carry out an ablation with various parts of our algorithm, showing the degradation in performance if we remove key parts. In all cases we use the structural Hamming distance (SHD) [56] to quantify how close a certain DAG is to another. The SHD measures how many insertions, deletions or flips are required to turn one graph into another. We also use the CPDAG SHD (SHD-C) [56] which computes SHD up to the equivalence class of completed partially directed acyclic graphs (CPDAG)s [42]. We give additional experimental details in the appendix B. In every case when we give the SHD produced by our model, this represents the SHD marginalized over our posterior, obtained by taking 100 samples of $W$ from

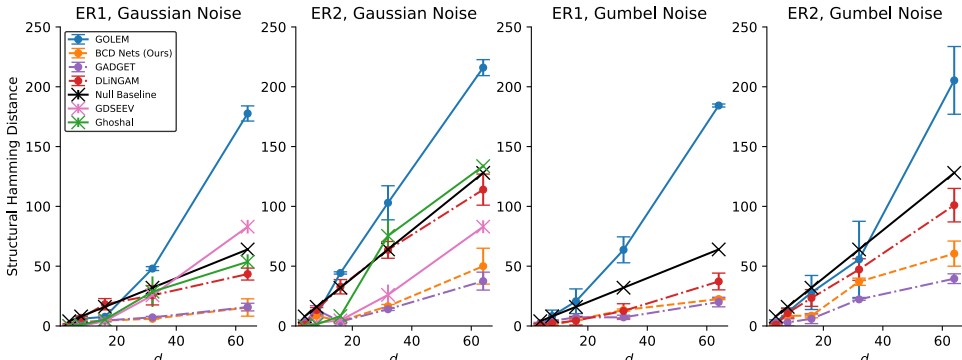

Figure 2: Distance between true graph and that estimated from BCD Nets, our variational approach, compared to several baselines described in the main text. *Lower is better.*

our converged posterior, computing the SHD and taking the average of these SHDs. Error bars are given by 5 random seeds for the data generation or choice of data subset.

## 5.1 Synthetic Data

We study our model's performance in the low-data regime, with $n = 100$ data points. We expect that when there is little data, the maximum-likelihood DAG may not be particularly representative of the DAGs that could have plausibly made the data, and so a Bayesian approach which can assign posterior density to multiple candidates may correctly identify more edges. In this experiment, we draw Erdős-Renyi graphs [14] with average degree equal to 1 or 2, denoted ER1 and ER2. The weights are generated uniformly in $\{-2, -0.5\} \cup \{0.5, 2\}$ following [35]. The data is generated in the equal-variance case so that the ground-truth is identifiable and the SHD is meaningful. We evaluate with Gaussian noise as well as with Gumbel noise to test how our method behaves under a misspecified likelihood. The distribution over the weights also induces some misspecification, since the weights are not drawn from the modelled prior of a horseshoe.

We compare against GOLEM [35], which attempts to find the maximum-likelihood solution of equation (1) using the continuous relaxation introduced in [62]. We found that the hyperparameters given by the authors did not lead to optimal performance with low data, so we found the best ones via a cross-validation scheme given in appendix B. We also include results from GADGET [58], a recent highly-optimized state-of-the-art Monte Carlo method for Bayesian causal discovery for DAGs, using 300,000 iterations with 48 parallel chains. To compare against a baseline that is widely used in practice, we also evaluate the DirectLiNGAM method [49], designed for non-Gaussian likelihoods. Also included are two methods designed specifically for Gaussian DAGs, the greedy directed acyclic graph search with equal error variance (GDSEEV) method [38], and the main method ('Ghoshal') from [21]. For the latter, we were unable to find an implementation by the authors and used our own implementation. We use the equal-variance formulations of GOLEM and BCD Nets.

Results are shown in figure 2. The null baseline is a method that simply predicts no edges. We observe that we reliably obtain significantly lower SHD than MLE approaches, especially at higher $d$ and degree where we tend to recover a significant fraction of the edges. For example, in the Gaussian noise case with average degree 2, for dimension 64 we get an SHD of $59 \pm 14$ compared to GOLEM's $205 \pm 30$. This corresponds to double the true positive rate while having a false discovery rate one-third of GOLEM's. We give full plots of the true positive rate, false positive rate and false discovery rates in the appendix, Section D.1. The Bayesian GADGET method also obtains better results than MLE methods, with BCD Nets' match to this exact sampling method suggesting that our approach finds a variational posterior close to the real posterior. Compared to GADGET, our method is generally faster (see table 2) and gives an explicit parametric form for the posterior, instead of simply giving samples. Furthermore, the ELBO is a natural metric to evaluate convergence, while for GADGET it is unclear how many iterations are required to mix[3]. DirectLiNGAM is competitive on the Gumbel likelihood, fitting its linear non-Gaussian framework.

---

[3] An earlier version of this paper reported worse SHDs for GADGET after using 50,000 steps. Increasing the number of iterations by 6 times resulted in much better mixing

## 5.2 Protein Dataset

We also evaluate on a benchmark protein signalling dataset [44]. The $d = 11$-dimensional dataset consists of $n = 853$ observations, with an expert-provided ground-truth graph. The true structure has 17 edges. We train on random draws of 100 observations. The results are shown in table 1. For GOLEM, both the equal-variance and non-equal variance maximum-likelihood methods perform worse than predicting no edges at all. Our equal variance method does not perform much better. Our non-equal variance method performs much better than maximum likelihood, obtaining an SHD of 15 from only 100 samples, close to the SHD of 14 obtained by GOLEM-NV when using the entire 853 data points (reported in Ng et al. [35]). NOTEARS, from [62] performs better than GOLEM. Meanwhile, the score-based GES method also achieves an SHD-C of 14, showing that our method approaches the performance of methods which enumerate all graphs (but scale badly). The non-probabilistic DLiNGAM method achieves similar SHD, although has the lowest SHD-C, indicating it gets some edges correct but the direction wrong. Finally, GADGET [58] achieves a slightly lower SHD of 13.9, within error of our approach and GES.

## 5.3 Causal Inference

To illustrate how our distributional approach could be used for causal inference, we test the ability of our method to make interventional predictions. In this experiment we generate a synthetic ER graph of degree 1 as in Section 5.1. We then choose a random edge in the graph between nodes $i, j$, with $x_i \rightarrow x_j$. Using the ground-truth parameters $W^*, \Sigma^*$, we choose a random value $a$ and sample $x_j \sim \mathrm{do}(x_i = a|W^*, \Sigma^*)$, by directly intervening in the data-generating process. For an estimated $\hat{W}, \hat{\Sigma}$ we can also sample $x_j \sim \mathrm{do}(x_i = a|\hat{W}, \hat{\Sigma})$. For

Table 1: Causal discovery approaches on the protein dataset with reduced data ($n = 100$)

|  | # Edges | SHD | SHD-C |
|---|---|---|---|
| GOLEM-EV | $1.5 \pm 1.3$ | $18.5 \pm 1.3$ | $18.5 \pm 1.3$ |
| GOLEM-NV | $1.5 \pm 1.3$ | $18.5 \pm 1.3$ | $18.5 \pm 1.3$ |
| NOTEARS | $18.5 \pm 0.8$ | $16.5 \pm 0.9$ | $17 \pm 1.0$ |
| GES | $12 \pm 0.9$ | — | $14.6 \pm 2.0$ |
| PC | $4.6 \pm 0.5$ | — | $14.0 \pm 0.6$ |
| GADGET | $4.7 \pm 0.5$ | $13.9 \pm 1.2$ | $13.7 \pm 0.6$ |
| DLiNGAM | $4.6 \pm 0.5$ | $14.8 \pm 1.0$ | $12.4 \pm 0.5$ |
| BCD Nets-EV | $11.3 \pm 1.2$ | $19.5 \pm 0.3$ | $19.4 \pm 0.1$ |
| BCD Nets-NV | $9.2 \pm 2.0$ | $14.7 \pm 0.9$ | $14.0 \pm 1.0$ |

our distributional approach, we marginalize over the final posterior distribution of parameters $q_\phi(W, \Sigma)$, drawing $x_j$ from the distribution with probability $\mathbb{E}_{W,\Sigma \sim q_\phi} [P(\mathrm{do}(x_i = a|W, \Sigma))]$. We then compare the sampled empirical distribution of $x_j$ to the ground truth interventional distribution. Our marginalization over the posterior allows us to get significantly closer to the true interventional distribution, measured by the Wasserstein distance (e.g. 0.25 for the linear-Gaussian graph at $d = 64$, compared to 2.8 for GOLEM). Full results as a function of $d$ are given in appendix E.1.

## 5.4 Model Running Time

We expect the time taken to train our model will asymptotically scale as $\mathcal{O}(d^3)$, similarly to maximum-likelihood methods [35, 62]. In table 2 we give the time taken to train to convergence as we vary the dimension $d$. The time taken for GOLEM includes the time required to choose the sparsity parameter $\lambda$ via cross-validation. BCD Nets didn't need any cross-validation to choose sparsity parameters. All the methods were run on the same hardware, a single Nvidia 2080Ti GPU with 16 CPUs. Our method takes more time to converge than GOLEM. This is not surprising, since we are training a neural network with many Sinkhorn iterations per optimization step. The Ghoshal algorithm [21] is very fast in comparison, consisting only of a precision estimation step then $d$ stages of matrix multiplication. At high $d$, GADGET is slower than our method. We found that speeding up GADGET by reducing the number of iterations resulted in dramatically reduced performance.

## 5.5 Increasing Dataset Size

We expect that our method performs best in the low-data regime, where a Bayesian approach with correctly-specified priors has an advantage over non-Bayesian methods. To show this, we perform experiments with increased amounts of data, results of which are shown in figure 3. We see that the advantage of BCD Nets is reduced as the quantity of data used is increased. We suspect that our variational method suffers from a more challenging optimization problem as the posterior becomes more peaked around the correct solution. Methods such as amortized inference or sequential Monte-Carlo may help to combat this problem and improve our model at larger $n$.

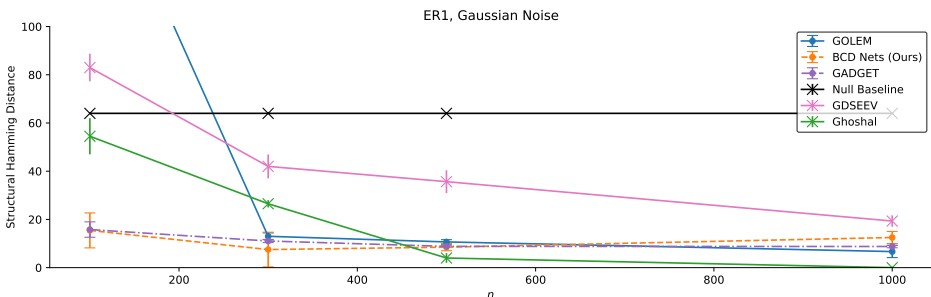

Figure 3: Structural Hamming Distance to the ground truth as a function of $n$, on 64-dimensional, degree 1 Erdős-Renyi graphs. With large amounts of data, baselines catch up to BCD Nets.

## 5.6 Ablation

Finally we perform an ablation of our algorithm to illustrate the importance of architectural choices. We use the $d = 32$ case, average degree 1, with Gaussian likelihood. We report the change in performance with a factored posterior (**Mean-Field**) and with a Laplace prior on the weights (**Laplace**) instead of a Horseshoe prior. We also report performance with a fixed 100 Sinkhorn steps (**Sinkhorn-100**), as

Table 2: The computing time required, in minutes, to converge to a solution for several approaches.

| $d$ | 8 | 16 | 32 | 64 | 128 |
|---|---|---|---|---|---|
| GOLEM | 25 | 30 | 40 | 65 | 90 |
| GADGET | 40 | 150 | 385 | 635 | 1200 |
| GHOSHAL | <1 | <1 | <1 | 3 | 15 |
| BCD Nets (Ours) | 50 | 160 | 300 | 350 | 900 |

opposed to the adaptive number that we use in our approach. We also report the KL-divergence between the sample covariance and the covariance induced by the sampled parameters $(\Sigma, L, P)$, which incorporates how well the posterior approximates the distribution of $L$, unlike SHD. We give the results in table 3 and 4, in the appendix. Changing any of the parts of the algorithm result in SHD increasing from 11 to around 30. Interestingly, under the Laplace prior the SHD is high but the KL-divergence is only modestly higher than with the Horseshoe. This indicates a learned DAG which can generate the data, but with extra edges due to an ineffective sparsity prior.

## 6 Conclusion

We introduce Bayesian Causal Discovery Nets, a framework for performing Bayesian causal discovery for linear-Gaussian structural equation models. BCD Nets exploit recent advances in variational inference to flexibly model the posterior distribution over SEM parameters given data, outperforming methods that only return point estimates. An explicit posterior is also useful in the high-stakes, low-data regimes where causal inference is increasingly used [57, 47].

On the other hand, while indications are that our method may be robust to a small amount of misspecification, the validity of the linear modelling assumption should be carefully considered in applications. Assuming a linear relationship is present where a highly non-linear one exists could lead to harmfully incorrect inferences, particularly for minority groups with complex and understudied variable interactions. Furthermore, our approach assumes that no *unobserved confounders* are present: variables which influence the observed variables but are not themselves observed. Since it is difficult to obtain useful inferential results under the effects of possibly arbitrary unknown confounders, the assumption of no unobserved confounders is standard in many areas of causal inference [38]. However, this assumption is unlikely to hold exactly in real applications, and so the possible influence of unobserved confounders must be seriously considered and their influence reduced as much as possible before using our method. Future work could explore improvements to inference speed by replacing the Sinkhorn with more efficient algorithms from optimal transport.

## Acknowledgements

This research was supported by NSF(#1651565, #1522054, #1733686), ONR (N000141912145), AFOSR (FA95501910024), ARO (W911NF-21-1-0125) and the Sloan Fellowship. We thank the anonymous reviewers for their helpful comments.

## Funding Transparency Statement

1. **Funding**: This research was supported by NSF(#1651565, #1522054, #1733686), ONR (N000141912145), AFOSR (FA95501910024), ARO (W911NF-21-1-0125) and the Sloan Fellowship

2. **Competing Interests**: There are no competing financial interests that could be perceived to influence the contents of this work.

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
