## A  Marginalizing Gaussian Edge Weights

If we set the prior over edge weights to be a $d(d-1)/2$-dimensional isotropic Gaussian with standard deviation $\nu$, so $l \sim \mathcal{N}(0, \Sigma_l)$ with $\Sigma_l = I\nu^2$, then we can analytically marginalize out the weights in the likelihood expression below. For an arbitrary datapoint $X$, we have

$$p(X|P, \Sigma) = C \int_{\mathbb{R}^{d(d-1)/2}} \sqrt{\det \Theta} \exp\left(-\frac{X^\top \Theta X + \nu^{-2} l^\top l}{2}\right) dl, \tag{4}$$

with $\Theta = (I - PLP^\top)^\top \operatorname{diag}(\sigma^{-2})(I - PLP^\top)$ as discussed above, and $L \in \mathbb{R}^{d \times d}$ a strictly lower triangular matrix with the $d(d-1)/2$ lower entries forming the vector $l$. Here, $C = (2\pi)^{-d(d-3)/4}\nu^{-d(d-1)/2}$ is a constant term. The following theorem provides a closed-form expression for this integral.

**Theorem 1.** *The log-likelihood* $\log P(X|P, \Sigma)$ *as defined in equation* (4) *is*

$$\log p(X|\Sigma, \nu, P) = -\frac{d}{2}\log 2\pi - \sum_{j=1}^{d} \log \sigma_j - (d-1)\log \nu + \frac{1}{2}X^\top(S - \Sigma)X - \frac{1}{2}\log \det D'$$

$$-\frac{1}{2}\log \det(\Sigma'^{-1} + D'^{-1}\nu'^{-2}), \tag{5}$$

*with* $D = \operatorname{diag}[PGP^\top X^2]$ *where $G$ is the strictly lower-triangular matrix with all lower-triangular entries equal to* $1$. *D has (almost surely) one zero diagonal entry, with index $i$. Primed matrices ($\Sigma'$, $D'$) are in $\mathbb{R}^{d-1 \times d-1}$, with the $i$th diagonal entry removed. Finally, $S \in \mathbb{R}^{d \times d}$ is a diagonal matrix with $S_{jj} = \frac{\nu^2 D_{jj}}{\sigma_j^2(\sigma_j^2 + \nu^2 D_{jj})}$.*

*Proof.* In the case where the edge weights are known to be drawn from a $d(d-1)/2$-dimensional isotropic Gaussian, $l \sim \mathcal{N}(0, I\nu^2)$, then we are able to analytically marginalize out the weights in the term $\mathbb{E}_{l \sim \mathcal{N}}[p(X|L, P, \Sigma)]$. We have

$$p(X|P, \Sigma, \nu) = \int_l p(X|L, P, \Sigma)P(L|\nu)dl.$$

We can compute this analytically. We write $l$ for the vector of $d(d-1)/2$ strictly lower-triangular elements, and $L$ for the matrix with these as the lower-triangular elements. We then have

$$p(X|P, \Sigma, \nu) = \int_l \frac{1}{(2\pi)^{d/2}(\det \Sigma)^{1/2}} \frac{1}{(2\pi)^{d(d-1)/4}\nu^{d(d-1)/2}} \exp\left(-\frac{1}{2}X^\top \Theta X\right) \exp\left(-\frac{1}{2\nu^2}l^\top l\right) dl.$$

assuming that the mean of $l$ and $X$ are zero. Now

$$\Theta = (I - W)\operatorname{diag}(\sigma^{-2})(I - W)^\top,$$

or

$$\Theta = (I - PLP^\top)^\top \operatorname{diag}(\sigma^{-2})(I - PLP^\top),$$

with

$$W^\top = PLP^\top,$$

(note that this is different from the $W = PLP^\top$ parameterisation in the main text, but is equivalent) and a vector of noise standard deviations $\sigma$. We have $\det \Theta = 1/\det \Sigma$ because $\det(I - PLP^\top) = \det(PIP^\top - PLP^\top) = \det P(I - L)P^\top = 1$.

Writing $\Sigma$ for the diagonal matrix $\operatorname{diag}(\sigma^2)$, if we expand out this matrix product we get $\Theta = \Sigma^{-1} - PL^\top P^\top \Sigma^{-1}PLP^\top - PL^\top P^\top \Sigma^{-1} - (PL^\top P^\top \Sigma^{-1})^\top$. Overall then, inside the exponential we have $X^\top \Theta X = X^\top \Sigma^{-1}X - X^\top PL^\top P^\top \Sigma^{-1}PLP^\top X - 2X^\top PL^\top P^\top \Sigma^{-1}X$. Since $L$ is the integration variable, the first term (not containing $L$) can be taken outside the integral.

We now note an interesting fact about the $X^\top PL^\top P^\top \Sigma^{-1} PLP^\top X - 2X^\top PL^\top P^\top \Sigma^{-1} X$ terms in the integral. Although we are integrating over the $d(d-1)/2$-dimensional space of $l$ values, this term is only $d$-dimensional. For example, in the $3 \times 3$ case, with identity permutation, we have

$$L = \begin{pmatrix} 0 & 0 & 0 \\ l_1 & 0 & 0 \\ l_2 & l_3 & 0 \end{pmatrix}.$$

and so

$$PLP^\top X = \begin{pmatrix} 0 & 0 & 0 \\ l_1 & 0 & 0 \\ l_2 & l_3 & 0 \end{pmatrix} \begin{pmatrix} x_1 \\ x_2 \\ x_3 \end{pmatrix} = \begin{pmatrix} 0 \\ l_1 x_1 \\ l_2 x_1 + l_3 x_2 \end{pmatrix},$$

where $x_1, \ldots, x_d$ are the $d$ individual components of the vector $X$. This means that if we are to do a change of variables into the basis

$$u = \begin{pmatrix} l_1 x_1 \\ l_2 x_1 + l_3 x_2 \\ l_2 x_1 - l_3 x_2 \end{pmatrix}$$

then along the third coordinate direction the $PLP^\top X$ terms will not change, since it's orthogonal to the terms involved in the integrand.

So we now change variables from $l$ to $u$, where the first $d-1$ components of $u$ are given by (the $d-1$ non-identically-zero components of) $PLP^\top X$, and the remaining $d(d-1)/2 - (d-1)$ components are vectors orthonormal to those directions. We write $u'$ for the $d-1$-dimensional vector formed from the nonzero components of $PLP^\top X$.

Now since the first components of $u$ are not unit length, we get a term in the $dl$ differential element corresponding to the length of the vectors, which would be e.g. $\left(x_1, \sqrt{x_1^2 + x_2^2}\right)$ in the case above. Given that we're substituting $u$ for $l$ we also will need to rescale the $\Sigma_l$ term.

We can express this via a diagonal matrix $D_{xP}$ which we can construct as

$$D = \operatorname{diag}\left(PGP^\top X^2\right),$$

where $G$ is the matrix with a $1$ in the strictly lower-triangular entries and $0$ otherwise.

We now have

$$C(X, P) \int_{u'} \exp\left(-\frac{1}{2}\left(u'^\top (\Sigma'^{-1} + D'^{-1}\nu^{-2})u' - 2u'\Sigma'^{-1}X'\right)\right) du',$$

with a constant $C(X, P) = \frac{1}{(2\pi)^{d/2}(\det \Sigma)^{1/2}} \frac{1}{(2\pi)^{(d-1)/2}\nu^{d-1}} \exp\left(-\frac{1}{2}X^\top \Sigma X\right) \frac{1}{(\det D')^{1/2}}$, and we have integrated out the $d(d-1)/2 - (d-1)$ dimensions of $u$ not involved in the term with $\Theta$. Primed matrices (e.g. $\Sigma' \in \mathbb{R}^{(d-1)\times(d-1)}$) have had the column and row corresponding to the zero entry of $D$ removed. We can then use the identity

$$\int_x \exp\left[-\frac{1}{2}\mathbf{x}^T \mathbf{A}\mathbf{x} + \mathbf{c}^T \mathbf{x}\right] d\mathbf{x} = \sqrt{\det\left(2\pi \mathbf{A}^{-1}\right)} \exp\left[\frac{1}{2}\mathbf{c}^T \mathbf{A}^{-T}\mathbf{c}\right]$$

where here we have $A = (\Sigma'^{-1} + D'^{-1}\nu^{-2})$ and $c = \Sigma'^{-1}x'$, we get

$$\int_{u'} \exp\left(-\frac{1}{2}\left(u'^\top (\Sigma'^{-1} + D'^{-1}\nu^{-2})u' - 2u'\Sigma'^{-1}X'\right)\right) du'$$

$$= (2\pi)^{(d-1)/2}\left(\det\left(\Sigma'^{-1} + D'^{-1}\nu^{-2}\right)\right)^{-1/2} \exp\left(\frac{1}{2}(\Sigma'^{-1}X')^\top (\Sigma'^{-1} + D^{-1}\nu^{-2})^{-1}\Sigma'^{-1}X'\right),$$

We end up with

$$\frac{1}{(2\pi)^{d/2}(\det \Sigma)^{1/2}} \frac{1}{\nu^{d-1}} \exp\left(-\frac{1}{2}X^\top \Sigma X\right) \frac{1}{(\det D')^{1/2}}$$

$$\times \left(\det\left(\Sigma'^{-1} + D'^{-1}\nu^{-2}\right)\right)^{-1/2} \exp\left(\frac{1}{2}(\Sigma'^{-1}X')^\top (\Sigma'^{-1} + D^{-1}\nu^{-2})^{-1}\Sigma'^{-1}X'\right).$$

We can also write the term $(\Sigma'^{-1}X')^\top(\Sigma'^{-1}+D^{-1}\nu^{-2})^{-1}\Sigma'^{-1}X'$ as $XSX^\top$ with a diagonal matrix $S$, where

$$S_{ii} = \frac{\nu^2 d_i}{\sigma_i^2(\sigma_i^2 + \nu^2 d_i)},$$

with $d_i$ the $i$th diagonal entry of $D$, so

$$p(X|\Sigma, \nu, P) = \frac{1}{(2\pi)^{d/2}\prod_i \sigma_i}\frac{1}{\nu^{d-1}}\exp\left(-\frac{1}{2}X^\top \Sigma X\right)\frac{1}{(\det D')^{1/2}}\left(\det\left(\Sigma'^{-1}+D'^{-1}\nu^{-2}\right)\right)^{-1/2}$$
$$\times \exp\left(\frac{1}{2}X^\top S X\right).$$

This gives a log-likelihood

$$\log p(X|\Sigma, \nu, P) = -\frac{d}{2}\log 2\pi - \sum_i \log \sigma_i - (d-1)\log \nu$$
$$-\frac{1}{2}X^\top \Sigma X - \frac{1}{2}\log \det D' - \frac{1}{2}\log\det(\Sigma'^{-1}+D'^{-1}\nu^{-2}) + \frac{1}{2}X^\top S X.$$

$\square$

## B Additional Experimental Details

**Baseline GOLEM settings:** As discussed in the main text, we found that the suggested [35] regularization hyperparameter of $\lambda_1 = 2 \cdot 10^{-2}$, led to poor performance. This poor performance at low $n$ is described in appendix G of [35]. However, we found that varying $\lambda_1$ could increase performance. We therefore hold out one-fifth of the data when training GOLEM, and use this as a validation set for cross-validation over $\lambda_1 \in \{2 \cdot 10^{-2}, 2 \cdot 10^{-1}, 2, 20\}$, scoring fit based on the sample mean-squared error $X - W^\top X$. GOLEM has better performance when trained for longer than the $10^5$ steps suggested, and so we trained for $2 \cdot 10^5$ steps, or until the gradient norm was lower than $0.01$. Additionally, we tested a probabilistic formulation of GOLEM by training an ensemble of GOLEM models on bootstrapped dataset. We used 20 models for the ensemble, excluding 5% of the dataset in each bootstrap dataset. We found this method was not competitive with the other baselines (e.g. SHD of 150 on the 32-dimensional ER2 graph), and didn't evaluate this baseline further due to its extremely large computational cost.

**Implementation details:** In all experiments we use a two-layer multilayer perceptron with 128 hidden units to parameterize the function returning the parameters $T$ of the Gumbel-Sinkhorn in $q(P|L, \Sigma)$. We use the adabelief optimizer [63] with a step size $10^{-3}$. We also apply a scaled sigmoid to the logits produced by the MLP so that they are in the range $(-20, 20)$ to increase stability of training. Nonetheless, the training can still be unstable at times, with sudden drops in the ELBO. During training, we found it useful to monitor the KL divergence between $\mathcal{N}(0, \hat{\Theta}^{-1})$ and $\mathcal{N}(0, \Theta_\phi^{-1})$, where $\hat{\Theta}$ is the empirical precision of the training data and $\Theta_\phi$ is the average precision matrix described in section 2.2, generated by our parameterized posterior distribution. We found that this quantity correlates strongly with the sample Hamming distance, and a sudden increase can indicate optimization issues during training. In the synthetic data cases, we used the code[4] provided by the authors of [35] to generate the data and compute the structural Hamming distance. Due to the large number of Sinkhorn steps required, we used the implicit derivative based on the adjoint method, provided by the optimal transport tools library[5]. This avoids the high memory cost of differentiating through the computation graph of the Sinkhorn iterates. We set $\tau$ for the

---

[4] https://github.com/ignavier/golem, Apache 2.0 license
[5] https://github.com/google-research/ott, Apache 2.0 license

horseshoe prior as $2/(d\sqrt{n})$ for all the experiments, since we expect average degree on the order of 1. The structural Hamming distance reported for our predictions, is computed via sampling from the posterior distribution and averaging the SHD induced by all samples.

**Sinkhorn Implementation Details:** To ensure we stick closely to the set of permutation matrices, we use a relatively small value of $\tau$, 0.2. We also use the straight-through gradient estimator [4]. This means that on the forward pass of the backpropagation algorithm, we use the $\tau \to 0$ limiting value of the Sinkhorn algorithm. On the backward pass the gradients are taken with respect to the finite-$\tau$ doubly-stochastic matrix. To obtain the $\tau = 0$ output we use the Hungarian algorithm [27]. Compared to previous work using Gumbel-Sinkhorn [31, 28], we note that we require a large number of Sinkhorn iterations for stable training, sometimes requiring over 1000 iterations to ensure the row and column sums equal 1 within a tolerance of 0.01.

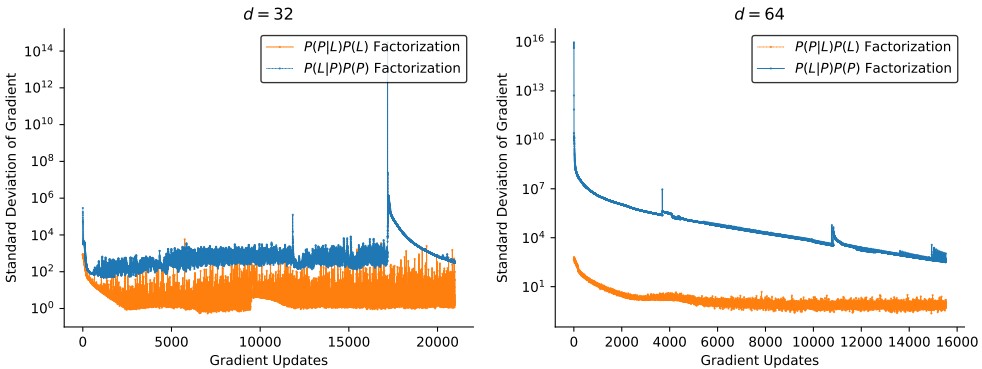

Figure 4: The variance of the gradient when training with two alternative factorizations. We note that the factorization where $P$ is sampled first has much higher gradient variance, leading to unstable and difficult optimization.

## C    Gradient Variance

In the main paper we discuss the possibility of an alternate factorization of the probabilistic model, $q_\phi(L, \Sigma|P)q(P)$ where we generate $P$ first, followed by $L$ and $\Sigma$. However, empirically we find that the factorization order in equation (3) performs much better than the alternative factorization, with orders of magnitude lower variance in the gradient (Figure 4). This suggests that the bias-variance trade-offs due to stochastic optimization in performing VI over $P$ (relaxations to discrete distribution over permutations, see Section 3.2.2) are more challenging than $(L, \Sigma)$ (continuous r.v. with de-facto pathwise gradient estimators, see Section 3.2.1).

## D    Additional Experimental Results

### D.1    Additional Prediction Statistics

In this section, we give the true positive rate, false positive rate, and false discovery rate of our algorithm and all the baselines. They are shown in figure 5. We observe that we typically perform better on all the metrics, with the exception of the false positive rate for the gumbel cases compared to DLiNGAM. As the DLiNGAM method is designed for the gumbel case while BCD Nets use a Gaussian likelihood, this makes sense.

### D.2    Qualitative Analysis

As a brief qualitative analysis the performance of our approach, we consider the three-dimensional case, where we are able to enumerate the $3! = 6$ permutation matrices. We draw a set of $n$ data points and train our model until convergence, in the equal-variance setting (ensuring a unique solution). We then draw one hundred samples from the joint distribution $q(P, L, \Sigma)$, and plot a histogram of the estimated distribution of $P$, shown in figure 6. As intuitively expected, the posterior is quite diffuse

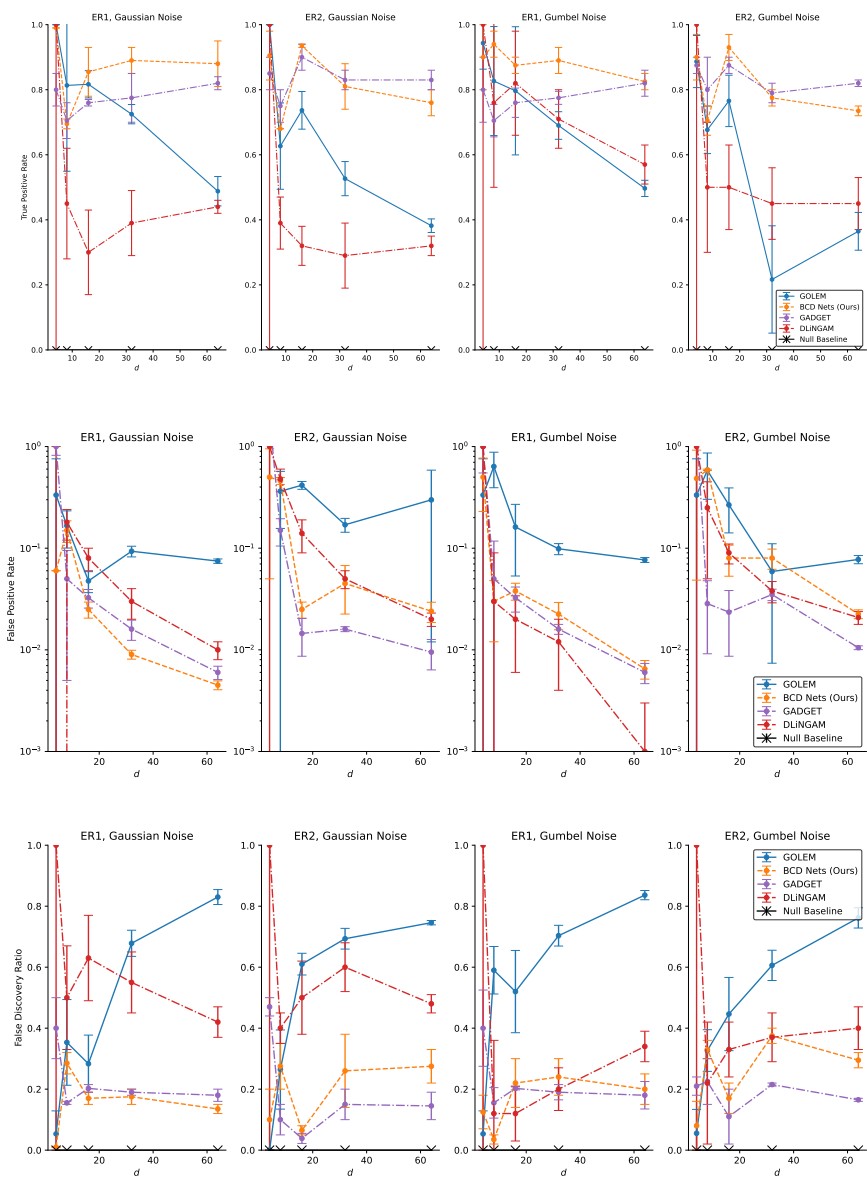

Figure 5: True positive rate, false positive rate, and false discovery rate for the experiments described in Section 5.1

for small amount of data, but quickly peaks around two possible candidate graphs. With additional data the ground-truth graph is picked out.

# E    Ablation Tables

Here we give the ablation results when we take away parts of the algorithm. The results are show in tables 3 and 4, demonstrating a degredation in performance when the architectural choices described in the main text are changed.

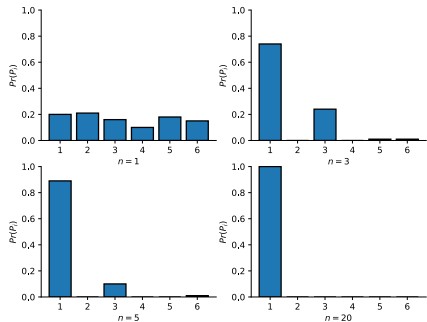

Figure 6: Concentration of the posterior distribution over the $3! = 6$ possible permutations for $n$ samples from a three-dimensional DAG, where the data-generating permutation is $P_1$. The posterior quickly concentrates onto two candidates, then onto the ground-truth permutation.

Table 3: Degradation of performance when removing components of our equal-variance model. Reducing the number of Sinkhorn iterations and using a mean-field posterior drastically degrade the solution quality. Using a Laplace prior instead of horseshoe modestly worsens performance w.r.t. KL divergence, but quite negatively impacts the SHD.

|  | SHD | Sample-KL |
|---|---|---|
| BCD Nets-EV | 11 $\pm1$ | $6.2 \pm 0.3$ |
| Mean-Field | $27 \pm 3$ | $21 \pm 2$ |
| Laplace | $34 \pm 1$ | $8 \pm 1$ |
| Sinkhorn-100 | $38.5 \pm0.5$ | $73 \pm 7$ |

## E.1 Causal Inference Experiment Results

Here we give full results of the causal inference experiment described in Section 5.3. Figure 7 shows the Wasserstein distance between the true interventional distribution and the estimated one, which we call the 'Estimated Intervention Distance'. This is averaged over random choices of the edge to intervene on, as well as over random seeds. We observe that we are able to effectively estimate effects of interventions. The degradation of performance in GOLEM likely arises due to the method incorrectly assigning edge weights at higher $d$ (see e.g. the much higher false positive rate of GOLEM compared to BCD Nets in figure 5) and so predicting a highly incorrect interventional distribution.

## E.2 Variation of $p$

In this section, we analyse the performance of the methods as the degree $p$ increases. As we expect, the performance of all the methods decreases substantially as the degree increases. The full results are shown in figure 8. We see that when the degree is 4, all the models do worse than chance. This is not too surprising as inferring the presence of 256 edges with only 100 data points is a very challenging task.

Table 4: Degradation of performance when removing components of our approach, for the non-equal-variance model. We see very similar behaviour to the equal-variance case.

|  | SHD | Sample-KL |
|---|---|---|
| BCD Nets-NV | $9 \pm 3$ | $7 \pm 3$ |
| Mean-Field | $33 \pm 1$ | $35.0 \pm 0.5$ |
| Laplace | $34.5 \pm 0.5$ | $9 \pm 1$ |
| Sinkhorn-100 | $31 \pm 1$ | $56 \pm 11$ |

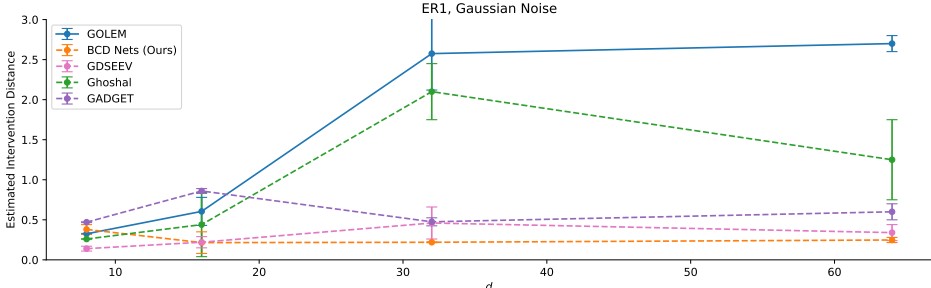

Figure 7: The average Wasserstein distance between the estimated distribution from an intervention and the true distribution from the intervention. Lower is better.

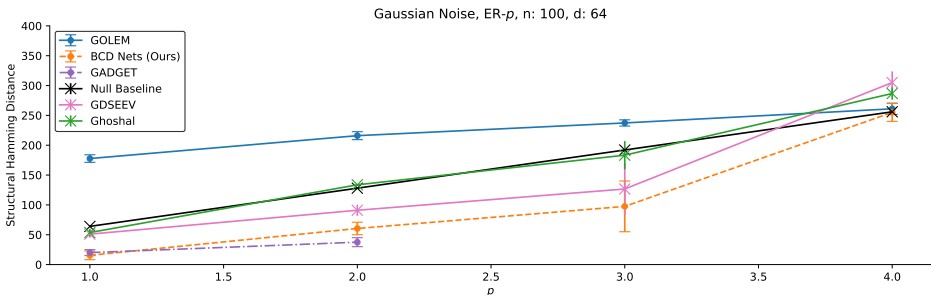

Figure 8: Structural Hamming Distance as a function of the degree $p$, for the inferred graph compared to the true graph, on random ER-1, 64-dimensional graphs with 100 samples. We see that with large enough degree, all the methods perform poorly.

## F Recovery of Ground Truth for Posterior

Here we discuss the asymptotic behavior of the posterior distribution for the SEM parameters under an infinite amount of data which has been generated from a linear Gaussian generative process with true parameters $\Sigma^*, W^*$. We sketch an argument for why we would recover the ground-truth parameters (or a certain equivalence class of the ground truth) under infinite data.

The posterior density for $\Sigma, W$ under an observation of $n$ data points $X_1^n$, is

$$P(\Sigma, W|X_1^n) \propto P(X_1^n|\Sigma, W)P(\Sigma, W), \qquad (6)$$

which we can write as a sum over data points

$$\log P(\Sigma, W|X_1^n) = \sum_i \log P(X_i|\Sigma, W) + \log P(\Sigma, W), \qquad (7)$$

where the likelihood is given by equation (1). We now argue that the posterior concentrates around the set of SEM parameters quasi-equivalent to the ground-truth parameters, for quasi-equivalence defined in [35]. We assume that the prior has support over the true parameters. Now, we assume that the posterior density concentrates around the set of maximum a posteriori (MAP) points. In [35], the problem

$$\mathrm{argmax}_{\Sigma, W \in \mathcal{W}} \left\{ \sum_i^n \log P(X_i|\Sigma, W) + n\lambda R_{\mathrm{sparse}}(W) \right\}, \qquad (8)$$

is studied, where $\mathcal{W}$ is the set of DAGs, $R_{\mathrm{sparse}}$ is a regularizer encouraging sparsity, $\lambda$ a chosen parameter, and $n \to \infty$. The likelihood is the same as ours. The solution is a set of $W$s which have corresponding DAGs $G$, which are quasi-equivalent to the true DAG $G^*$. The corresponding edge weights and noise variances are simply a (regularized) linear regression problem when conditioned on the graph structure, so have a Gaussian likelihood given $G$. The conditions on $\lambda$ in [35] are not

particularly well defined, only requiring "weights for regularization terms such that the likelihood term dominates asymptotically".

In order to match equation (8) with equation (7) and show that the MAP points of our distribution are quasi-equivalent to the true parameters, we would have to use a sparsity-encouraging prior and assume that we could choose $\lambda = 1/n$. However, this would mean the likelihood grows while the regularizer stays constant. By analogy to the BIC score [46], we might want to choose a $\lambda$ such that the regularization term grows modestly with $n$. We could achieve this in our setting by choosing a prior parameterized by $n$, such as a Laplace or Horseshoe prior with scale proportional to $n^{-1/2}$.

Finally, we note that in [38], in the proof of theorem 1 it is directly shown that in the equal variance case, the solution to the problem in equation (8) is uniquely the ground truth set of parameters. So under the equal variance assumption and suitable sparsity priors, the posterior will concentrate to the ground truth parameters.

## G   Computational Use

While developing this work, we estimate we used a total of around 1000 GPU-hours on a Nvidia 2080Ti, on an internal cluster, leading to emissions of around 100kg CO2 equivalent [6].

---

[6]Using https://mlco2.github.io/impact