# OpenReview forum: "BCD Nets: Scalable Variational Approaches for Bayesian Causal Discovery"
_NeurIPS.cc/2021/Conference — NeurIPS 2021 Poster_

### Official Review · Reviewer_nWpC · 2021-07-02

**Rating:** 6
**Confidence:** 4

**Summary:**

The paper proposes a variational inference approach for approximate Bayesian causal discovery. In that, it focuses on modelling distribution over linear-Gaussian (LG) SEMs. The LG-SEMs are represented as tuples of an observation noise matrix  $\Sigma$ and a weighted adjacency matrix $W = PLP^\top$  and, which are modelled as combination of strictly lower-triangular real-valued matrix $L$ followed by a variable permutation represented by permutation matrix $P$. The proposed algorithm maximizes the ELBO w.r.t. to the parameters $\phi$ of the variational posterior $q_\phi(P|L, \Sigma) = q_\phi(P|L, \Sigma) q_\phi(L, \Sigma)$. Here, $q_\phi(P|L, \Sigma)$ is modelled as conditional Bolzmann distribution where re-parametrized sampling of permutation matrices $P$ is facilitated via the Gumble-Sinkhorn relaxation - this way pathwise gradient estimators of the ELBO can be obtained. In the experiment section, the proposed method is compared to three related approaches on simulated Erdos-Renyi graphs and on the well-known protein-signalling task.

**Limitations And Societal Impact:**

The paper discusses the limitations of the linear Gaussian SEM assumption. It would be good to also discuss how restrictive the chosen variational family $\{q_\phi(P, L, \Sigma): \phi \in \Phi\}$ is.

Since the paper concerns algorithmic aspects of approximate Bayesian causal discovery, it is generally applicable in many fields - thus, I do no expect any immediate societal impact.

**Main Review:**

**Writing:** The paper is well-written, and very easy to follow. Though, some parts of the paper are a bit verbose, particularly Section 2, 3.3 and 5. Making them more concise would free up space that could be used to e.g. strengthen the experiment section.

**Relevance:** The paper addresses a relevant problem of facilitating Bayesian inference over SCMs. The key motivation for a Bayesian treatment of causal discovery is to be able to quantify the epistemic uncertainty over SCMs - in principle the proposed method is able to do so.

Unfortunately, the paper hardly evaluates the uncertainty quantification aspect of the proposed method. The TP and FP rate are displayed in the appendix. Why not report the AUROC as done by most papers in the field of Bayesian Causal Discovery? Then the information is compressed into a single number and you can present the results in the main paper.

Also Fig. 5 in the appendix is a nice result - but why doesn’t it include GADGET and DLiNGAM? I recommend adding the corresponding results for the other two methods and moving the result to the main paper. Alternatively, you could report the interventional log-likelihood as done in [3] - this nicely evaluates the methods towards the purpose of reasoning about interventions under uncertainty - the main purpose of Bayesian causal discovery.

**Originality & Significance:** None of the technical pieces in the paper is particular novel / original. However, the paper skilfully assembles recent method such as the  re-parameterized sampling of permutation matrices via the Gumbel-Sinkhorn distribution, the approximation of the matrix permanent, the horseshoe prior to enforce sparsity etc., to develop a tractable variational inference algorithm over SCMs. Thus, the overall algorithm can certainly be considered a contribution to the research field. A particular advantage of the method is that it does not need to explicitly enforce a acyclicity constraint like the NOTEARS method [1].

**Quality & soundness:** The proposed method is sound and well-motivated. Unfortunately, the experiment section is a bit weak. Two of the three baseline methods do not even model a distribution over DAGs - both output a point estimate. The only really relevant baseline is GADGET. It would be good to include simple baselines that can capture epistemic uncertainty such as bootstrapped versions of standard algorithms such as LiNGAM, GES or NOTEARS (e.g. see [2]). Moreover, the ER-1 graphs used in the experiment section are very sparse. It would be good to include an experiment with harder problems such as ER-3 or ER-4 where predicting the null graph is not already a strong baseline.

A major concern of mine is the lack of code which makes the method & experiment results very hard to reproduce. Since this paper is a resubmission and there is more than a week after the deadline to prepare and polish the supplementary material, failing to provide usable and well-documented code is definitely a red flag.

**Overall Assessment:** All in all, the method proposed in the paper is relevant, sound, well-presented and certainly of interest to the causal inference community - not a ‘game changer’ but a solid contribution.  Unfortunately, the paper falls short of a convincing empirical evaluation and fails to provide code. In the current form I do not see the paper above the acceptance threshold for NeurIPs. However, if the authors sufficiently address the mentioned shortcomings I am happy to change my assessment.

[1] Xun Zheng, Bryon Aragam, Pradeep Ravikumar, and Eric P. Xing. 2018. “DAGs with NO TEARS: continuous optimization for structure learning”, NeurIPS, 2018.
[2] Agrawal, Raj, et al. "Abcd-strategy: Budgeted experimental design for targeted causal structure discovery.", AISTATS, 2019.
[3] Lorch, Lars, et al. "DiBS: Differentiable Bayesian Structure Learning." arXiv preprint arXiv:2105.11839 (2021).


**Time Spent Reviewing:**

4-5 hours

---

> ### Author Response · Authors · 2021-08-10
> **Reply to Reviewer nWpC**
>
> Thank you for your thorough review. We address your specific concerns below:
>
> `Why not report the AUROC as done by most papers in the field of Bayesian Causal Discovery?`
>
> Thank you for the excellent suggestions about reporting the AUROC.  We struggled to fit in the TPR, FPR, FDR plots, and we will include this in the main paper as suggested in the next revision.
>
> `Also Fig. 5 in the appendix is a nice result - but why doesn’t it include GADGET and DLiNGAM? I recommend adding the corresponding results for the other two methods and moving the result to the main paper.`
>
> We have updated figure 5 with the other baselines suggested, as well as with other baselines suggested by reviewers, [available at this URL](https://www.dropbox.com/s/jfzrncs8rl7lrib/new_eid_fig.png?dl=0). In the next revision we will include this in the main body. We see that both GDSEEV from [1] and the method from [2] improve on the performance of GOLEM, but that our method still obtains the most accurate interventional distribution. GADGET also improves on GOLEM but does not outperform LiGa-VI.
>
> `The only really relevant baseline is GADGET. It would be good to include simple baselines that can capture epistemic uncertainty such as bootstrapped versions of standard algorithms such as LiNGAM, GES or NOTEARS (e.g. see [2]).`
>
> Thank you for this suggestion, we have implemented a bootstrapped version of GOLEM, with results shown [at this URL](https://www.dropbox.com/s/4p4d6m9ba1jrijq/new_fig_1.png?dl=0). Due to time constraints (computing a 10-way bootstrapped Golem for d=32 required 20 hours of computation) we only evaluated the bootstrapped GOLEM for the degree 2 case, up to d=32. However, we observe that this method performs worse on SHD than the original GOLEM method. This is likely due to a phenomenon where a bad bootstrap draw of data leads GOLEM to predict a very incorrect adjacency matrix, worsening the average SHD.
>
> `Moreover, the ER-1 graphs used in the experiment section are very sparse. It would be good to include an experiment with harder problems such as ER-3 or ER-4 where predicting the null graph is not already a strong baseline.`
>
> We  performed experiments at higher degrees, ER3 and ER4, with d=64, with results [at this URL](https://www.dropbox.com/s/879shy594pbkpid/new_fig_high_p.png?dl=0). We observe that our method continues to outperform the baselines in the ER3 case, but that in the very challenging ER4 case all the baselines (and LiGa-VI) do not significantly outperform predicting the null graph.
>
> `A major concern of mine is the lack of code which makes the method & experiment results very hard to reproduce.`
>
> We have released the code [at this link](https://www.dropbox.com/sh/ezd52k87aabfqao/AADIy8lR2wZuNGVatwnR-FmJa?dl=0)  and we will open-source it upon publication.

---

> > ### Comment · Reviewer_nWpC · 2021-08-20
> > **Reply**
> >
> > Thank you for providing further experiment results and code. As promised, I will raise my score from 5 to 6.

---

### Official Review · Reviewer_MkjZ · 2021-07-15

**Rating:** 7
**Confidence:** 3

**Summary:**

This submission presents an approach to quantify the uncertainty in estimating linear DAG structure using variational inference. In particular, the authors present a set of modeling design choices that (i) lead to tractable gradients of the ELBO, and (ii) are robust to certain classes of model misspecification. They go on to demonstrate that their approach outperforms maximum likelihood-based approaches on synthetic and semi-synthetic data.

**Ethical Concerns:**

None.

**Limitations And Societal Impact:**

No, these authors have not considered limitations and potential negative societal impact of their work. That being said, the methodological focus of this submission means that this is no more a concern than any other work.

**Main Review:**

Overall this submission addresses an important and often overlooked task of quantifying uncertainty in structure learning. As the authors note, quantifying uncertainty in this way is particularly important in settings where the number of observations are limited, or when structure can not be uniquely identified from observational data asymptotically. In addition to the problem being relevant, the design decisions are reasonable and well motivated.

However, I believe the submission could be significantly improved by (i) expanding the baselines and metrics in the empirical evaluation and (ii) clarifying the necessary assumptions relating the proposed approach to causality.

Empirical evaluation:

As the authors note, probabilistic approaches to structure learning have many benefits over maximum likelihood approaches. However, the authors do not evaluate their claim that variational approaches provide scalability benefits over e.g. sampling-based approaches to probabilistic structure learning. As one of several example baselines to compare against, Mohammadi and Wit, 2015 propose a birth-death MCMC algorithm for a very similar (or identical) problem formulation of linear Gaussian DAGs. In addition to this (potentially strong) baseline, it would be good to compare against a more naive sampling-based approach such as a random walk MCMC over graph permutations and weights.

In addition to new baselines, the experiments could be improved by evaluating the performance at various observational sample sizes for identifiable graphs. My expectation is that we should see the maximum likelihood approaches and the variational approaches converge to the same performance as n gets very large. This would emphasize the domain under which the proposed method is an appropriate choice, which I believe is large d and small n.

Despite these critiques, I am very happy to see that the authors performed an ablation study of their proposed method. This honest empirical assessment is a very good trend for the machine learning research community.

Clarity:

As the authors claim that their method can "infer causal relationships" it is extremely important to specify the necessary assumptions for this interpretation upfront. As an example, are the authors assuming causal sufficiency (i.e. no latent confounders)?

Minor critique:

There are many spelling and grammatical errors throughout the draft that negatively impact its quality. I highly recommend taking a thorough editorial pass through future versions of the submission. This has not affected my recommendation.

A. Mohammadi, E. C. Wit. Bayesian Structure Learning in Sparse Gaussian Graphical Models. Bayesian Anal. 2015

Update: I have increased my score after seeing the authors' response.

**Time Spent Reviewing:**

3

---

> ### Author Response · Authors · 2021-08-10
> **Reply to Reviewer MkjZ**
>
> Thank you for your thorough review. We address your specific concerns here:
>
> `the submission could be significantly improved by (i) expanding the baselines and metrics in the empirical evaluation`
>
> Following up on the reviewer suggestion, we have evaluated against several new baselines presented by all the reviewers, and extended the regimes at which we compare to higher degree and dataset size. The additional baselines can be seen [at this URL](https://www.dropbox.com/s/4p4d6m9ba1jrijq/new_fig_1.png?dl=0), where we incorporate methods from [1] and [2], as well as a bootstrapped GOLEM. The new Gaussian methods are more competitive than GOLEM at low-$d$, but at high $d=64$ our method has the lowest SHD. We also evaluate on ER-3 and ER-4 graphs, in the Gaussian setting. We give a plot of our results [at this URL](https://www.dropbox.com/s/879shy594pbkpid/new_fig_high_p.png?dl=0). In this setting, all the methods degrade in performance quite heavily, although LiGa-VI still has the lowest SHD.
>
> `Mohammadi and Wit, 2015 propose a birth-death MCMC algorithm for a very similar (or identical) problem formulation of linear Gaussian DAGs. In addition to this (potentially strong) baseline, it would be good to compare against a more naive sampling-based approach such as a random walk MCMC over graph permutations and weights.`
>
> Thank you for bringing our attention to this previous work. If time permits, we will try to adapt this work as one of our baselines to compare against. However, as far as we can tell, this work is in the setting of structure learning in undirected graphs, which is quite a different setting to ours. For the comparison to the sampling approach, we believe our comparison to GADGET (a state-of-the-art sampling-based approach, Neurips 2020) shows that we can outperform advanced sampling based methods.
>
> `the experiments could be improved by evaluating the performance at various observational sample sizes for identifiable graphs. My expectation is that we should see the maximum likelihood approaches and the variational approaches converge to the same performance as n gets very large`
>
> We agree that this exploration of behavior with $n$ would be useful. We agree that from a theoretical point of view as $n$ increases we should see the maximum likelihood approaches and Bayesian approaches converge (in identifiable cases). We discuss this in more detail in our appendix, section G.
>
> Experimentally, we have carried this out for the Gaussian case [with results at this URL](https://www.dropbox.com/s/clzi1tzp0jgdwr0/new_fig_n.png?dl=0), where we can see that the behavior follows your intuitions. Both GDSEEV and Ghoshal improve in performance as $n$ increases. At $n=1000$, Ghoshal narrowly outperforms LiGa-VI, likely due to the higher $n$ leading to a more sharply peaked posterior and a worse-conditioned optimization problem for LiGa-VI.
>
> [1] J. Peters and P. Bühlmann. Identifiability of Gaussian structural equation models with equal error variances. Biometrika, 101(1): 219--228, 2014.
>
> [2] Ghoshal A, Honorio J. Learning linear structural equation models in polynomial time and sample complexity. International Conference on Artificial Intelligence and Statistics. PMLR, 2018: 1466-1475. Besides,

---

> > ### Comment · Reviewer_MkjZ · 2021-08-18
> > **Thanks to the authors for the additional experiments**
> >
> > Thank you to the authors for responding to my concerns about the empirical evaluation. My apologies for missing that Mohammadi and Wit propose a method for undirected graphs.
> >
> > Given the updated experiments, I will revise my score accordingly.

---

### Official Review · Reviewer_PQX5 · 2021-07-16

**Rating:** 6
**Confidence:** 2

**Summary:**

This paper introduces a causal discovery method for linear Gaussian SEM. The authors extend the variational inference to learn the causal structure for linear SEM data.

**Limitations And Societal Impact:**

Yes

**Main Review:**

The paper considers a fundamental problem of learning causal relationships in the linear model with gaussian distributions.
The paper builds on prior work in the fields of variational inference and bayesian estimation. \
The authors put some effort into learning the posterior distribution $p(W,\Sigma|X)$ by variational inference.\
The paper provides the details of the critical distribution choices for the estimation of the parameters. \
The experiments are quite thorough, and the paper is well-organized.

However, there are some concerns or suggestions:

 -It is unfair to compare with the DLiNGAM. This method was designed for the non-Gaussian dataset. The authors may need to compare with methods that are designed for linear Gaussian data, such as [1,2]

1. J. Peters and P. Bühlmann. Identifiability of Gaussian structural equation models with equal error variances. Biometrika, 101(1): 219--228, 2014.

2. Ghoshal A, Honorio J. Learning linear structural equation models in polynomial time and sample complexity. International Conference on Artificial Intelligence and Statistics. PMLR, 2018: 1466-1475.
Besides,

-The algorithm is evaluated on synthetic cases with only 100 datapoints. It would have been more convincing with more data points.

-Why the PC and GES algorithms are not presented in the synthetic data? What is the reason?

**Time Spent Reviewing:**

5 hours

---

> ### Author Response · Authors · 2021-08-10
> **Reply to Reviewer PQX5**
>
> Thank you for your thorough and informative review.
>
> `The authors may need to compare with methods that are designed for linear Gaussian data, such as [1,2]`
>
> Thanks for bringing these additional baselines to our attention.  We have implemented these, leading to an updated figure 1 that can be viewed [at this URL](https://www.dropbox.com/s/4p4d6m9ba1jrijq/new_fig_1.png?dl=0). We see that GDSEEV from [1] and the algorithm from [2] (‘Ghoshal’) both have quite good performance up to d=32, but worsen in the high-dimensional  d=64 case, where our method obtains the lowest SHD.
>
> `Why the PC and GES algorithms are not presented in the synthetic data? What is the reason?`
>
> For figure 1, we wanted to compare directed methods, i.e. methods which return a specific DAG instead of a Markov equivalence class. As PC and GES only return a CPDAG, it’s not appropriate to compare them to directed methods on the basis of SHD. It is possible to compare them to other methods on the basis of SHD C, which operates on CPDAGs. A SHD_C version of figure 1 is similar to the SHD version, with GES performing slightly worse than other baselines on SHD_C and PC performing slightly better compared to other baselines. For example, for an ER-1 64-dimensional graph, the SHD_Cs are: GES: $105 \pm 15$, PC: $30 \pm 4$, DLiNGAM: $50 \pm 7$,LiGa-VI: $15 \pm 5$, Ghoshal: $55 \pm 10$.
>
> `The algorithm is evaluated on synthetic cases with only 100 datapoints. It would have been more convincing with more data points.`
>
> Our aim with this method is to develop an approach that works in the low-n regime, where it is an advantage to consider a distribution over possible DAGs that could have generated the data. As expected in other contexts, we expect that the advantages of a Bayesian distributional approach will diminish as the amount of data increases and the solution tends towards the MAP solution. Our preliminary results on higher $n$ **[here](https://www.dropbox.com/s/clzi1tzp0jgdwr0/new_fig_n.png?dl=0)** show this, with convergence between MAP methods and our distributional approach at higher n. Both GDSEEV and Ghoshal improve in performance as $n$ increases. At $n=1000$, Ghoshal narrowly outperforms LiGa-VI, likely due to the higher $n$ leading to a more sharply peaked posterior and a worse-conditioned optimization problem for LiGa-VI.

---

> > ### Comment · Reviewer_PQX5 · 2021-08-22
> > **Feedback after author response**
> >
> > Thanks a lot for your reply. Based on the author's response and the authors addressing the concerns noted by all other reviewers, I have increased my score to a 6.

---

### Official Review · Reviewer_uT5D · 2021-07-31

**Rating:** 8
**Confidence:** 4

**Summary:**

### 1. Summary

The paper proposes a novel framework in the field of (causal) structure learning, using a variational inference method.

This framework is proposed to be able to provide estimates of distribtutions in the graph regime of the DAG to be learned.


**Limitations And Societal Impact:**

I would suggest an additional paragraph about potential negative societal impact

**Main Review:**

### 2. Rationale for the score

The problem chosen is a well researched problem. Showing that indeed state of the art methods can be outperformed is remarkable.

The use of synthetic and real world data for experiments corresponds with the usual approach in this field of research.

The paper is well structured, graphical contributions in the main part as well as in the appendix serve to enable a good understanding of the approach done.

The novelty of providing distributions instead of point estimates is a good contribution in this field of research.

### 3. Positive aspects

- Using the Structural Hamming Distance is a good choice here to be able to compare true and learned DAGs

- The ablations performed are well chosen and provide a good benefit to the experiments done

- The CO2 impact mentioned in the appendix is a nice detail

- The "causal inference" section is a nicely chosen addition to the structure learning parts.


### 4. negative aspects

- I would have expected a codebase provided in the supplementing material for the implementation of the algorithm

- More discussion of the results and outlooks regarding special corner cases (e.g. hidden confounders (=unobserved variables) in the true graphs) would be a nice addition

**Time Spent Reviewing:**

5

---

> ### Author Response · Authors · 2021-08-10
> **Reply to Reviewer uT5D**
>
> Thank you for your careful review. We hope the following addresses your concerns:
>
> `I would have expected a codebase provided in the supplementing material for the implementation of the algorithm`
>
> We have included the code after some cleaning up, [available at this URL](https://www.dropbox.com/sh/ezd52k87aabfqao/AADIy8lR2wZuNGVatwnR-FmJa?dl=0), and will open-source it on publication.
>
> `More discussion of the results and outlooks regarding special corner cases (e.g. hidden confounders (=unobserved variables) in the true graphs) would be a nice addition`
>
> We agree that more discussion of the potential corner cases and limitations of this method is appropriate. We will expand the brief discussion of limitations in the conclusion (where we only say “the validity of the linear modelling assumption should be carefully considered in applications”) to include an explicit discussion of both the possible presence of latent confounders and how our model will not be appropriate in that setting.
>
> In particular, the presence of unobserved confounders violates our assumption that all the relevant variables are observed. This is a common assumption in causal discovery [1, 2]. In practice we would recommend leveraging the learned posterior distribution with the use of posterior predictive checks [3], comparing samples from the posterior to the actual data, to help catch cases where the data generating assumptions do not match the real generative process.
>
> [1] J. Peters and P. Bühlmann. Identifiability of Gaussian structural equation models with equal error variances. Biometrika, 101(1): 219--228, 2014.
>
> [2] Glymour, Clark, Kun Zhang, and Peter Spirtes. "Review of causal discovery methods based on graphical models." _Frontiers in genetics_ 10 (2019): 524.
>
> [3] Gelman, Andrew, et al. Bayesian data analysis. Chapman and Hall/CRC, 1995

---

> > ### Comment · Reviewer_uT5D · 2021-09-11
> > **I thank the authors for providing the codebase**
> >
> > I thank the authors for providing the codebase and am glad they look forward to open source the contribution. I also appreciate the elaborate answer to my comment. I agree that the the assumption of not having unobserved variables is quite common in the relevant literature. Expanding the discussion of limitations is an appropriate step here.
> >
> > As my original review is already ratimg the paper at an 8 which is quite high, I will keep the rating.

---

### Author Response · Authors · 2021-08-10
**Short Summary of Changes due to Reviews**

We thank all the reviewers for their comprehensive and thoughtful reviews. Several reviewers indicated that not including our code was a shortcoming. We have cleaned up the code and it is available (in anonymised form) [at this URL](https://www.dropbox.com/sh/ezd52k87aabfqao/AADIy8lR2wZuNGVatwnR-FmJa?dl=0).

Several reviewers pointed out that the experiments section could be strengthened with additional baselines and evaluations. We have carried out these additional experiments, which are listed as follows:

+ Additional baselines, namely from [1], [2]  as well as a bootstrapped GOLEM. These are evaluated in the same setting as figure 1 in our paper. The graph showing the results may be found [at this URL](https://www.dropbox.com/s/4p4d6m9ba1jrijq/new_fig_1.png?dl=0). Our method outperforms these baselines.
+ Additional baselines evaluated on figure 5. We include the GADGET, LiNGAM, [Peters 2014], [Ghoshal 2018], and a bootstrapped baseline suggested by the reviewers. The graph can be found [at this URL](https://www.dropbox.com/s/jfzrncs8rl7lrib/new_eid_fig.png?dl=0). These new baselines are more competitive with our approach but LiGa-VI narrowly outperforms them.
+ Higher values of $p$, the average degree. We evaluate on ER-3 and ER-4 graphs, in the Gaussian setting. The graph can be found [at this URL](https://www.dropbox.com/s/879shy594pbkpid/new_fig_high_p.png?dl=0). These show that all methods decrease in accuracy as the degree increases, although our method decreases the least.
+ Higher values of $n$, the dataset size. We evaluate on $n=(100,300,500,1000)$, showing the convergence of our methods to the maximum likelihood methods as $n$ increases and the posterior density concentrates to the MAP solution. The results can be seen [at this URL](https://www.dropbox.com/s/clzi1tzp0jgdwr0/new_fig_n.png?dl=0). Here we see the maximum likelihood methods improve in SHD as $n$ increases, converging to the performance of our method.

We address specific concerns to each reviewer in indivdualized comments.


[1] J. Peters and P. Bühlmann. Identifiability of Gaussian structural equation models with equal error variances. Biometrika, 101(1): 219--228, 2014.

[2] Ghoshal A, Honorio J. Learning linear structural equation models in polynomial time and sample complexity. International Conference on Artificial Intelligence and Statistics. PMLR, 2018: 1466-1475.

---

### Decision · Program_Chairs · 2021-09-27

**Decision:**

Accept (Poster)

**Comment:**

This article introduces an approach to causal graph inference using variational Bayesian inference.  The paper focuses on linear-Gaussian structural equation models, and employs a carefully designed variational approximation to the posterior on directed acyclic graphs (DAGs) to handle the challenging combinatorial nature of the graph space; the proposed approach is called LiGa-VI.  The posterior is parametrized in terms of $(P,L,\Sigma)$ where $P$ is a permutation matrix for the vertex ordering, $L$ is a lower triangular matrix such that $P L P^\texttt{T}$ is the graph adjacency matrix, and $\Sigma$ is a diagonal covariance matrix for the Gaussian model.  For the variational approximation $q_\phi(P,L,\Sigma)$, the authors use a normal distribution or normalizing flow for $q_\phi(L,\Sigma)$ and a Gumbel-Sinkhorn relaxation of the Gumbel-Matching distribution over permutations for $q_\phi(P|L,\Sigma)$. Gradient-based optimization is used to optimize the variational parameter $\phi$ to fit the posterior by minimizing KL divergence.  Experiments are performed to assess performance versus several competing methods on synthetic and real data.

The variational approximation used over DAG space is interesting and innovative.  The experimental performance appears to be surprisingly good in terms of concentration near the true graph.  Overall, the reviewers were quite positive about the paper, but I have some significant concerns.

1) No improvement in performance as the sample size $n$ increases? I'm puzzled by the performance as a function of the sample size $n$ in the additional figure supplied by the authors in their reply to Reviewer PQX5 (https://www.dropbox.com/s/clzi1tzp0jgdwr0/new_fig_n.png?dl=0).  Why does the proposed method (LiGa-VI) not benefit from having more data?  Similarly, why does the Bayesian method GADGET not benefit from having more data after $n$ increases beyond $300$?  Other methods such as Ghosal's are clearly benefiting from having more data.  This makes me concerned that the good performance of LiGa-VI here is solely due to something trivial like the choice of prior.  This needs to be explained.  Also, I highly recommend adding this plot to the paper.

2) I'm skeptical about Figure 2.  To have such a big difference in performance, it seems like something about this simulation is cherry picked to yield good performance.  Any Bayesian method with the same prior should yield roughly the same accuracy here, up to the accuracy of the posterior approximation.  Is there a big difference in the priors used by GADGET and LiGa-VI?  Or is GADGET not sampling well from the posterior?  It is fine to present simulations that showcase the proposed method, but they need to be put in context to show the relative strengths *and* weaknesses of the method.

3) The computation time of LiGa-VI is not that attractive (Appendix H, Table 4), taking around 2-10x the time required for GOLEM even though GOLEM is using cross-validation while LiGa-VI is not.  I found this disappointing since I would have expected a variational approach to be more computationally attractive.  I would highly recommend including the computation time for the other competing approaches, especially GADGET since it employs MCMC for Bayesian inference.

4) Mathematical notation and unclear exposition. Some of the notation and descriptions were not at all clear, such as "$q_\phi(L,\sigma) = \mathcal{N}(\mu_\phi,\sigma_\phi)$" for the variational approximation to the posterior on $(L,\Sigma)$ --- the left-hand side is a density while the right-hand side is a distribution, where does $L$ enter into the right-hand side, and how does this yield a distribution over $\sigma$?  Please make sure all mathematical expressions are properly written for clear exposition. I was also confused by this phrase in Appendix H: "we are training a neural network with many Sinkhorn iterations per optimization step", since I didn't see where LiGa-VI is using a neural network in the description of the method.  This needs to be clarified.